

# Characterization of indigenous *Durio* species from Sarawak, Borneo: relationships between chemical composition and sensory attributes

Gerevieve Bangi Sujang[1], Shiamala Devi Ramaiya[1], Shiou Yih Lee[2] and Muta Harah Zakaria[3]

[1] Department of Crop Science, Faculty of Agricultural and Forestry Sciences, Universiti Putra Malaysia Bintulu Campus, Bintulu, Sarawak, Malaysia
[2] Faculty of Health and Life Sciences, INTI International University, Nilai, Negeri Sembilan, Malaysia
[3] Department of Aquaculture, Faculty of Agriculture, Universiti Putra Malaysia, UPM Serdang, Selangor Darul Ehsan, Malaysia

Corresponding author
Shiamala Devi Ramaiya,
shiamala@upm.edu.my

## ABSTRACT

Sarawak, Borneo, harbours 16 unique *Durio* species, half of which are edible, with only *Durio zibethinus* widely cultivated. Despite their nutritional and economic significance to the rural communities in Sarawak, the lesser-known indigenous durians remain underrepresented in the scientific literature while facing the risk of extinction in the wild. Thus, the aim of this study was to conduct comprehensive chemical analyses of these wild edible durians, offering insights into their nutritional and sensory taste attributes. The edible part was separated at optimal ripeness, and the samples were subjected to further analysis. Wild edible durian genotypes exhibit varied characteristics, even within the same species. The majority of wild durians are characterized by a sugar composition consisting predominantly of sucrose, constituting 67.38–96.96%, except for the red-fleshed *Durio graveolens* renowned for its low total sugar content (0.49 ± 0.17 g per 100 g). Despite its bland taste, this species possessed significantly greater fat (14.50 ± 0.16%) and fibre (12.30 ± 0.14%) content. *Durio dulcis* exhibited a significantly greater carbohydrate content (29.37–30.60%), and its intense smell was attributed to its low protein content (2.03–2.04%). Indigenous durians offer substantial percentages of daily mineral intake, with 100 g servings providing approximately 15.71–26.80% of potassium, 71.72–86.52% of phosphorus, 9.33–27.31% of magnesium, and sufficient trace minerals. The vibrant flesh colours of yellow-, orange- and red-fleshed *Durio graveolens* and *Durio kutejensis* show high levels of ascorbic acid (31.41–61.56 mg 100 g$^{-1}$), carotenoids (976.36–2627.18 μg 100 g$^{-1}$) and antioxidant properties, while *Durio dulcis* and *Durio oxleyanus*, despite their dull flesh, contained high phenolic (67.95–74.77 mg GAE 100 g$^{-1}$) and flavonoid (8.71–13.81 QE mg 100 g$^{-1}$) levels. These endeavours provide a deeper understanding of the nutritional richness of wild edible durians, thereby supporting commercialization and conservation efforts.

# INTRODUCTION

Borneo is a treasure trove of biodiversity, and Sarawak, as a part of it, is known for its tropical rainforests with a variety of fruit-bearing trees. Durian, a *Durio s*p. crowned as the "king of fruits", is native to the tropics and Borneo as its center of distribution. It gained popularity by captivating the taste buds of individuals across various corners of the globe, especially in Asia, with its unique tropical delight aroma and taste. There are 16 *Durio* species present in Sarawak, half of which are edible; *Durio dulcis, Durio graveolens, Durio grandiflorus¸ Durio kutejensis, Durio lowianus, Durio testudinarum, Durio oxleyanus* and *Durio zibethinus* and the other half which are inedible; *Durio acutifolius, Durio carinatus, Durio lanceolatus, Durio griffithi, Durio affinis, Durio crassipes*, and *Durio excelsus* and each of these species has various vernacular or local names that vary across regions and ethnic groups in Sarawak (*Idris, 2011*). The widely cultivated and commercialized durian is primarily *Durio zibethinus*.

Wild edible durians are a valuable component of the biological richness that permeates Sarawak, Borneo's tropical ecology and landscape. However, these indigenous durians are still collected from their wildly grown such as *D. dulcis, D. graveolens, D. testudinarum*, and *D. oxleyanus*, while *D. kutejensis* is semi-cultivated (*Sujang et al., 2023*). These wild edible durians are popular among local people as they are frequently consumed by local communities and readily available in common local markets (the morning market, *Tamu* market, and wet market) during fruiting season. Compared with the commercially available *D. zibethinus* variety, these wild edible durians are unique in terms of size and aesthetic appearance. They have long sharp thorns, their flesh colour (ranging from yellow to orange to red), pericarp colour (ranging from yellow to green), and even unique taste. *Durio dulcis* has aesthetic value because of its attractive red pericarp colour, while the flesh of *D. graveolens* varies in colour, ranging from pale yellow to deep yellow, orange and crimson red (*Lim, 2012*). A study on the nutrient composition of *D. graveolens, D. dulcis*, and D. *kutejensis* revealed that these durians are nutritious and similar to other durian varieties (*Hoe & Siong, 1999*; *Belgis et al., 2016*; *Voon et al., 2007*; *Susi, 2017*). *Durio graveolens* has an intense yellowish‒orange‒to‒yellow pulp, indicating that these species contain acceptable amounts of carotenoids and fatty acids and are potential antioxidant fruits (*Voon et al., 2007*; *Khoo et al., 2011*; *Nasaruddin, Noor & Mamat, 2013*). These wild durian fruits are often pricier than commercially grown ones due to their availability and seasonality, with prices ranging from 20 MYR (Malaysian Ringgit) for four to five small fruits to 35 to 70 MYR per kilogram during the off season. The presence of these resources in rural areas not only signifies a potential source of nutrition but also suggests the possibility of them serving as a means of income for local communities. However, the potential of wild edible durian species has not been widely explored.

Despite much research on common *D. zibethinus* cultivars, little information is available for wild edible durians, as some of them are in tropical forests and either at risk of extinction or have already vanished (*Tan et al., 2020*). The scarcity of raw materials could be a significant obstacle hindering efforts to promote product commercialization. These wild edible durians may contain significant nutrients, bioactive compounds, and antioxidant capacity. Their antioxidant capacity may be comparable or superior to that of the more extensively studied *D. zibethinus* cultivars. These wild edible durians represent an opportunity for local growers to access special markets where consumers appreciate the exotic character of such products and the presence of bioactive compounds that promote potential health benefits. Therefore, in this study we performed a detailed chemical analysis of the wild edible durians, providing information into their nutritional composition, physical properties, chemical characteristics and sensory taste attributes. The study's significance lies in its potential to address the nutritional value of wild edible durians in order to encourage widespread consumption and acceptance of these fruits. In addition an aim was to aids in selecting valuable species for long-term biodiversity conservation efforts, and the cultivation of improved varieties contributing to economic opportunities.

## MATERIALS AND METHODS

### Sample collection

The *Durio* species used in this study were discovered from the northeast to central regions of Sarawak, Borneo, which consists of Bekenu, Niah, Baram, Sibu, and Tatau. Sampling was carried out from August 2021 to March 2022, the durian season. Table 1 shows the coordinates of each location marked with a Garmin GPS Handheld (GPSMAP79S, Olathe, KS, USA). Permission was granted by Sarawak Biodiversity Centre under reference number JKM/SPU/608-8/2/1 Vol. 3 to conduct research on indigenous durians from Sarawak

### Sample preparation

Naturally dropped, ripened fruits were selected and collected. Only fruits free of visual defects were chosen for analyses. For *D. graveolens*, the selection criteria focused on minimal splitting, as the fruits tend to split open on the tree when ripe. Additionally, fruits that produced hollow sounds when tapped were selected because this sound indicates ripeness (*Kharamat, Wongsaisuwan & Wattanamongkhol, 2020*). The collected durian fruits were transported to the laboratory. Durian fruits were dehusked along natural sections using a knife, except for *Durio dulcis*, which needed to be cut into halves using a machete due to its difficulty in opening. The durian pod was removed from the rind, and then the arils were separated from the seed manually and weighed for analysis. Fresh samples were used for moisture, ascorbic acid and sensory studies. For the oven-dried and freeze-dried samples, the durian pulp was dried in an oven (MEMMERT UFB500 Universal Oven, Schwabach, Germany) at 60 °C until a constant weight was obtained for 3 days, while for freeze-dried sample using a freeze dryer (SP Virtis 4KBTZL Benchtop

**Table 1** GPS coordinates and details of sampling locations for wild edible *Durio* species in Sarawak, Borneo.

| Code | Species | | Local name | Locations | Coordinates | Description |
|------|---------|---|------------|-----------|-------------|-------------|
| D1 | *D. dulcis* | | Tutong | Ulu Anap, Tatau | N2° 45′ 47.88″ E112° 57′ 43.92″ | Big-sized |
| D2 | *D. dulcis* | | Tutong | Ulu Anap, Tatau | N2° 45′ 47.88″ E112° 57′ 43.92″ | Small-sized |
| D3 | *D. graveolens* | | Isu Kuning | Teku, Sibu | N2° 17′ 16.00″ E111° 49′ 50.99″ | Yellow-fleshed |
| D4 | *D. graveolens* | | Isu Oren | Niah, Miri | N3° 51′ 43.20″ E113° 42′ 51.48″ | Orange-fleshed |
| D5 | *D. graveolens* | | Isu Merah | Long Bedian, Baram | N3° 51′ 43.20″ E113° 42′ 51.48″ | Red-fleshed |
| D6 | *D. kutejensis* | | Nyekak | Bekenu, Miri | N4° 3′ 29.47″ E113° 50′ 39.09″ | – |
| D7 | *D. oxleyanus* | | Daun | Niah, Miri | N4° 3′ 29.47″ E113° 50′ 39.09″ | Pale yellow-fleshed |
| D8 | *D. oxleyanus* | | Daun | Long Bedian, Baram | N3° 51′ 43.20″ E113° 42′ 51.48″ | Yellow-fleshed |
| D9 | *D. zibethinus* | | Terung iban | Ulu Anap, Tatau | N2° 45′ 47.86″ E112° 57′ 41.90″ | – |

Freeze Dryer, Cridersville, Ohio, US) for 48 h. The dried samples was homogenized to a fine powder and stored in airtight containers and a −20 °C freezer for oven-dried and freeze-dried samples, respectively.

## Determination of nutritional properties

### Physicochemical properties of wild edible Durio species

In the method described by *Tan et al. (2020)*, total soluble solid (TSS) was determined by blending 5 g of durian pulp with 10 mL distilled water and filtered with Whatman No.1 filter paper. Then, the filtrate was measured using a hand-held refractometer (ATAGO Corp., Tokyo, Japan), and the findings were given in ˚Brix (method 983.17, *Association of Official Agricultural Chemists (AOAC), 2000*). Total Titratable Acidity (TTA) was determined as a percentage of citric acid using the AOAC method 942.15. A 10 mL juice sample with six drops of 1% phenolphthalein indicator was titrated with 0.1N NaOH until a pink endpoint. The pH of the pulp was determined by using a pH meter (Mettler-Toledo, Greifensee, Switzerland).

The method of *Shaffiq et al. (2013)* was used to determine the sugar contents (fructose, glucose, sucrose, and maltose) using high-performance liquid chromatography (HPLC) (Waters Alliance 2695; Waters, Millford, MA, USA). To extract the samples, one gram of dried durian pulp was mixed with 10 mL of 80% methanol and heated in a water bath (MEMMERT WNB 45 Waterbath, Schwabach, Germany) at 80˚ for 60 min. After extraction, the solutions were centrifuged (Beckman Coulter J26 High Speed Centrifuge, Pasadena, CA, USA) at 2,500 × g for 20 min. The collected supernatants were then heated to 80˚ in a water bath and evaporated until dry to remove all ethanol content. The analytical column that was used is the the Sugar-Pak I column (Waters, Milford, MA, USA). The injected volume was 20 μL. At a flow rate of 0.4 mL per minute, the mobile phase (0.001 M Ca-EDTA in ultra-pure water) was pumped through the column. The sample identification and quantification of sugars were performed using an external standard method, where commercial standards obtained from Sigma for sucrose, glucose, fructose, and maltose were utilized. The sum of all individual sugar was reported as total sugars.

### Proximate analyses

Standard protocols of the *Association of Official Agricultural Chemists (AOAC) (2000)* were followed to determine the proximate composition. For the determination of moisture content, oven (MEMMERT UFB500 Universal Oven, Schwabach, Germany) drying (method 934.06) was used to obtain a consistent weight. Two hundred milligrams of dried durian pulp was incinerated at 550˚ for 5 to 6 h in a muffle furnace to estimate the ash content (method 930.05). Crude fat was determined using the Soxhlet extraction method (920.39) by using Foss Tecator Soxtec 2055 Manual Extraction. The crude fiber was estimated by acid-base digestion method (993.19) using 2010 Fibertec System Foss Tecator, Sweden. Protein content was determined using the Kjeldahl method (method 955.04). The carbohydrate content was determined through the difference method, subtracting the sum of moisture, protein, fat, and ash from 100.

### Mineral compositions

The obtained ash was used to extract the minerals using the double acid method. The extracted solution was transferred to a volumetric flask, and the volume was increased to

100 mL. The solution was then used for mineral content determination. The standard protocol of the *Association of Official Agricultural Chemists (AOAC) (2000)* (method 975.03) was followed for the determination of calcium (Ca), potassium (K), sodium (Na), magnesium (Mg), iron (Fe), zinc (Zn), manganese (Mn) and copper (Cu) using atomic absorption spectrum (AAS) (AA800 Perkin-Elmer, Germany). Phosphorus (P) was determined by colorimetric method using a UV-VIS spectrophotometer (*Murphy & Riley, 1962*) at a wavelength of 882 nm.

### Vitamins

The carotenoid content was analyzed following *Tan et al. (2020)* method. Initially, 15 g of durian pulp was homogenized, and 25 mL of acetone was put in a separatory funnel containing 40 mL of petroleum ether. The acetone was then gradually removed from the samples using deionized water. The aqueous phase of the extract was released, and this process was repeated two times until no residual solvent remained. The extract was mixed with 15 g of anhydrous sodium sulphate in a 50 mL volumetric flask and the volume was made up of petroleum ether. The samples were examined at 450 nm with a UV-Vis spectrophotometer, and the formula for calculating the total carotenoid content of the durian pulp:

$$\text{Carotenoids content (kg/kg)} = \frac{absorbance \ \times volume \ of \ extract \ (ml) \times 104}{2{,}592 \ \times sample \ weight \ (kg)}$$

where 2,592 is the beta-carotene extinction coefficient in petroleum ether.

The *Association of Official Agricultural Chemists (AOAC) (2000)* indophenol titration method 974.29, was used to determine vitamin C content in the durian samples, where ascorbic acid as standard (1 mg mL$^{-1}$) was used. Briefly, 10 g of durian pulp was homogenized with 90 mL of distilled and filtered through Whatman No. 2 filter paper. Two (2 mL) of the filtrates were added with 5 mL of acid stabilization solution (15 g of metaphosphoric acid dissolved in 40 mL glacial acetic acid, then diluted to 500 mL with distilled water), then immediately titrated with 2, 6 dichlorophenolindophenol solution until the pink colour persists for 15 s. The ascorbic acid content was calculated according to the following equation:

$$\text{Vitamin C (mg/100 g)} = \frac{Ts \ \times Vstd \ \times Cstd}{Tstd \ \times Vs} \times 100$$

where $Ts$ is titrant for sample (mL, titrant for sample-blank), $Vstd$ is a volume of the standard used (mL), $Cstd$ is a concentration of the standard (1 mg mL$^{-1}$), $Tstd$ is titrant for standard (mL, titrant for standard-blank) and $Vs$ is a volume of sample used (mL).

### Phytochemical analyses

The 5 g freeze-dried and powdered durian pulp was dissolved in 250 mL of 80% methanol extract for three days at room temperature and placed in an orbital shaker. The resulting extracts were stored at −18˚ until further analysis. Total phenolic content (TPC) was analyzed by the reaction with the Folin-Ciocalteu reagent. The absorbance was measured using a UV-Vis spectrophotometer at 765 nm. Quantification was performed based on the calibration curve of gallic acid (0–50 mg L$^{-1}$), and the results were expressed in mg gallic

acid equivalent (GAE) 100 $g^{-1}$ sample. Additionally, total flavonoid content (TFC) was determined by using UV-Vis Spectrophotometer at the wavelength of 510 nm. The absorbance was measured against the blank at 510 nm with an UV-Vis spectrophotometer (Lambda 25, Perkin Elmer, Shelton, CT, USA). A calibration curve was constructed using standard quercetin, and the total flavonoid content was expressed as mg of quercetin (QE) equivalents per 100 $g^{-1}$ sample. Meanwhile, the Ferric Reducing Antioxidant Power (FRAP) assay measures the ability of the antioxidants in the investigated samples to reduce ferric-tripiridyltriazine ($Fe^{3+}$ TPTZ) to a ferrous form ($Fe^{2+}$). Briefly, 1 mL of sample extract was mixed with 2.8 mL of ferric-TPTZ and incubated at 37° for 10 min. The mixture's absorbance was measured at 593 nm using a UV-Vis spectrophotometer 2,2-Diphenyl-1-picrylhydrazyl (DPPH) free radical scavenging activity of durian was measured by the methods described by *Wang & Li (2011)*. One (1 mL) of DPPH ethanolic solution (0.1 mM) was added to 0.5 mL of various concentrations from 20–100 µg $mL^{-1}$ of samples, and the absorbance at 519 nm was measured on a UV-Vis spectrophotometer with 95% methanol used as the blank.

## Sensory analysis

Fifteen trained panellists underwent sensory assessment using the Quantitative Descriptive Analysis (QDA®) following ISO Standard 8586-2. As decribed by *Voon et al. (2007)* after two screening rounds to assess sensory abilities, the trained panellists received four sessions of descriptive analysis training over 4 weeks. Sessions lasting 1–2 h to ensure the evaluators could accurately describing sensory aspects. Panellists evaluated nine durian samples for attributes (Table 2) like appearance, flavour, and taste using a sensory score sheet with 15 cm unstructured scale lines (0–15). Palate were cleared with mineral water and unsalted crackers between samples, with short breaks for fresh air intake to prevent sensory fatigue and ensure consistent evaluations.

## Statistical analyses

The mean, standard deviation, and range of physicochemical attributes of the wild edible durians were calculated based on three replicate measurements. The statistical software SAS window program 9.4 was used to calculate the means and standard errors in any analysis. The statistical significance of any observed differences was evaluated using single-factor ANOVA, using the *Post-hoc* Tukey's ($p < 0.05$). The $EC_{50}$ values for DPPH were calculated by linear regression analysis. For sensory attributes, mean data on the descriptive analysis by the panellist were analyzed using Spider Web in Excel version 2013. Principal component analysis (PCA) was performed using XLSTAT software (Addinsoft, Paris, France) on sensory attributes and physicochemical components to study the interrelationship among the different attributes.

## RESULTS

### Physicochemical composition of wild edible *Durio* species

Physicochemical factors, such as sweetness and consumer satisfaction, are pivotal for ensuring optimal quality and influencing fruit taste (*Nunes et al., 2016*). There were

**Table 2  Sensory attributes used for quantitative descriptive analysis of minimally processed durian.**

| Attributes | Description |
|---|---|
| **Appearance** | |
| Brightness | The appealing and vivid colour of the durian flesh, is reminiscent of the vibrant hues found in fruits like oranges and dragon fruit. |
| Dullness | The lack of intensity in the colour pigment of the durian flesh, a comparable colour found in honeydew and bottle gourd. |
| **Flavour** | |
| Sweetness | The taste is akin to the flavour of sugar. It reflects the sweet sensation experienced when consuming sugar solutions. |
| Bitterness | The taste resembles the bitterness found in coffee, tea or burned food. |
| **Taste** | |
| Creaminess | Refers to a high viscosity texture that is reminiscent of whipped cream and, a smooth and rich mouthfeel. |
| Smoothness | This attribute indicates a texture devoid of roughness, often experienced when consuming items like bananas and avocados. |
| Juiciness | The amount of juice or moisture perceived during chewing is akin to the moist sensation experienced when eating fruit like grapes. |
| Stickiness | This texture relates to a sticky sensation experienced in the mouth, similar to the texture of sticky candy. |
| **Overall aftertaste** | Refers to the lingering sensory experience that occurs once the initial taste of the durian flesh has been consumed, which persists in the mouth and nasal passages after swallowing. |

**Table 3  The physicochemical properties of wild edible *Durio* genotypes.**

| Genotypes | | pH | TTA (%) | TSS (°Brix) | Total sugar (g 100 g$^{-1}$) |
|---|---|---|---|---|---|
| D1 | *D. dulcis* | $6.19 \pm 0.02^{ef}$ (6.17 − 6.22) | $0.20 \pm 0.04^{bc}$ (0.13 − 0.26) | $27.33 \pm 0.44^{a}$ (26.50 − 28.00) | $21.90 \pm 0.09^{b}$ (21.73 − 22.05) |
| D2 | *D. dulcis* | $6.12 \pm 0.02^{f}$ (6.10 − 6.15) | $0.27 \pm 0.03^{b}$ (0.19 − 0.29) | $28.00 \pm 1.04^{a}$ (26.00 − 29.50) | $22.20 \pm 0.06^{b}$ (22.08 − 22.30) |
| D3 | *D. graveolens* (yellow-fleshed) | $6.89 \pm 0.02^{a}$ (6.85 − 6.92) | $0.39 \pm 0.01^{a}$ (0.38 − 0.42) | $21.17 \pm 0.88^{b}$ (19.50 − 22.50) | $9.67 \pm 0.10^{f}$ (9.49 − 9.84) |
| D4 | *D. graveolens* (orange-fleshed) | $6.31 \pm 0.01^{d}$ (6.28 − 6.32) | $0.14 \pm 0.01^{cd}$ (0.13 − 0.16) | $21.50 \pm 0.58^{b}$ (20.50 − 22.50) | $17.57 \pm 0.32^{c}$ (17.12 − 18.18) |
| D5 | *D. graveolens* (red-fleshed) | $6.23 \pm 0.01^{e}$ (6.20 − 6.25) | $0.14 \pm 0.01^{cd}$ (0.13 − 0.16) | $12.50 \pm 0.29^{c}$ (12.00 − 13.00) | $0.49 \pm 0.17^{g}$ (0.40 − 0.56) |
| D6 | *D. kutejensis* | $6.42 \pm 0.01^{c}$ (6.41 − 6.44) | $0.11 \pm 0.01^{cd}$ (0.10 − 0.13) | $27.17 \pm 0.44^{a}$ (26.50 − 27.00) | $16.38 \pm 0.15^{d}$ (16.12 − 16.63) |
| D7 | *D. oxleyanus* | $6.75 \pm 0.00^{b}$ (6.75 − 6.76) | $0.11 \pm 0.01^{cd}$ (0.10 − 0.13) | $22.83 \pm 1.30^{b}$ (20.50 − 25.00) | $13.71 \pm 0.05^{e}$ (13.66 − 13.81) |
| D8 | *D. oxleyanus* | $6.86 \pm 0.01^{a}$ (6.84 − 6.88) | $0.14 \pm 0.01^{cd}$ (0.13 − 0.16) | $21.17 \pm 0.60^{b}$ (20.00 − 22.00) | $13.59 \pm 0.02^{e}$ (13.56 − 13.63) |
| D9 | *D. zibethinus* | $6.42 \pm 0.01^{c}$ (6.40 − 6.43) | $0.09 \pm 0.01^{d}$ (0.06 − 0.10) | $25.00 \pm 0.50^{ab}$ (24.00 − 25.50) | $27.66 \pm 0.05^{a}$ (27.58 − 27.66) |

**Note:**
Data presented are means ± standard error (SE) from three replications ($n$ = 3). Different superscript alphabets in the same column indicate differences at $p < 0.05$ (ANOVA, Tukey's test). Abbreviations: TTA, total titratable acidity; TSS, total soluble solid.

significant differences ($p < 0.05$) among the nine wild edible *Durio* species in terms of all physicochemical characteristics tested, as presented in Table 3. The pH of the wild edible *Durio* genotypes was slightly acidic, ranging from 6.12 ± 0.02 in *D. dulcis* (D2) to 6.89 ± 0.02 in yellow-fleshed *D. graveolens* (D3). *Durio dulcis* has a lower pH, ranging from 6.12 to 6.19, due to its greater total titratable acidity compared to the other genotypes. The pH values observed in this study fell within the reported range of values found in *D. kutejensis* (batuah) from Indonesia (6.87) (*Belgis et al., 2016*) and *D. zibethinus* (D101) from Malaysia (6.88) (*Voon et al., 2007*). Analysis of variance in total titratable acidity (TTA) was also significantly different ($p < 0.05$) among the genotypes. The TTA contents of the wild edible *Durio* genotypes were within the range of 0.09 ± 0.01% for *D. zibethinus* (D9)

to 0.39 ± 0.01% for yellow-fleshed *D. graveolens* (D3). All the genotypes were within the range of *D. zibethinus* among the Malaysian cultivars (0.09% of which were Chuk and 0.26% of which were MDUR78) reported by *Voon et al. (2007)*, except for yellow-fleshed *D. graveolens* (D3), which was twice higher than the other genotypes. The pH and TTA influence the organoleptic quality of durian and contribute to its overall sensory perception and taste profile (*Esti et al., 2002*).

Sugars constitute the primary soluble element found in fruit and are important for determining their taste and flavour (*Yang et al., 2021*). Total soluble solids (TSS) and total sugar were significantly different ($p < 0.05$) among the *Durio* genotypes. The TSS among the genotypes ranged from 21.17 ± 0.88 to 28.00 ± 1.04 °Brix, except for the red-fleshed *D. graveolens* (D5), which had the twice lowest TSS value at 12.50 ± 0.29 °Brix. The TSS of the present study was lower than that of the Malaysian variety *D. zibethinus* (29.47–34.57 °Brix) (*Tan et al., 2020*). The °Brix value of *D. kutejensis* (D6) was similar to the TSS range reported by *Belgis et al. (2016)*, who studied lai (*D. kutejensis*) Indonesia cultivars. The variation in TSS content in wild edible *Durio* genotypes contributed to the variability in sugar content (*Kader, 2008*). The total sugar content of the wild edible *Durio* genotypes ranged from 0.49 to 27.66 g 100 g$^{-1}$. The total sugar content of *D. zibethinus* (D9) was 56 times higher than that of red-fleshed *D. graveolens* (D5). To enhance the bland taste of the red-fleshed *D. graveolens*, also known as 'is merah' (Sarawak) or 'durian merah' (Sabah), Sabah local people make it into *tempoyak* (*Belgis et al., 2016*). Typically, locals prefer to consume fruit that are rich in fructose and glucose in their raw state, while fruit with lower sugar content are often utilized in cooking or consumed in combination with other ingredients (*Shaffiq et al., 2013*). The total sugar content of *D. kutejensis* (D6) was within the range of lai (*D. kutejensis*) cultivars (11.70 to 18.95 g 100 g$^{-1}$) reported by *Belgis et al. (2016)*.

The predominant sugar in durian was sucrose, followed by glucose, fructose, and maltose (*Aziz & Jalil, 2019*). Figure 1 shows the significant differences in sucrose, glucose, fructose, and maltose content among the nine wild edible *Durio* genotypes. Compared with fructose, glucose, and maltose, sucrose accounted for 67.38 to 96.96% of the total sugar composition in the different genotypes, similar to the findings of previous studies by *Voon et al. (2007)* and *Tan et al. (2020)*, except in red-fleshed *D. graveolens* (D5), where 52.08% of the sugar was predominant by fructose. *Durio zibethinus* (D9) had significantly ($p < 0.05$) higher sucrose (22.98 g 100 g$^{-1}$) and maltose (2.65 g 100 g$^{-1}$) concentrations, which were consistent with the trend of the °Brix readings (Table 3). The high sugar content in *D. zibethinus* (D9) reflects its high carbohydrate content. The sucrose concentrations observed in the wild edible *Durio* genotypes in this study were approximately half the reported levels found in *D. zibethinus* varieties (*Charoenkiatkul, Thiyajai & Judprasong, 2016*), owing to variations in enzyme activity across different cultivars and species (*Lee et al., 2013*).

## Proximate composition of wild edible *Durio* genotypes

The results of the proximate properties of the *Durio* species are shown in Table 4. The proximate properties of the nine wild edible *Durio* genotypes were significantly different

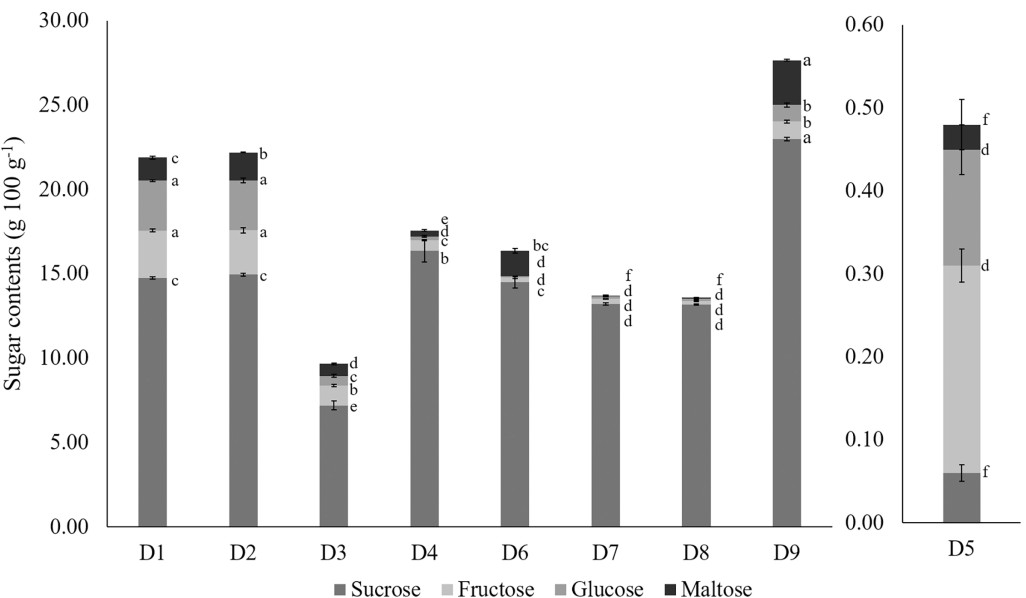

**Figure 1** **The sugar composition of wild edible *Durio* genotypes.** The error bars above each bar in the graph represent the standard error of the mean (*n* = 3). Different subscripts alphabets indicate differences at *p* < 0.05 (ANOVA, Tukey's test). The code of genotypes can be referred to Table 1.

**Table 4 Proximate composition of wild edible *Durio* genotypes.**

| GE | Moisture (%)* | Ash (%) | Fiber (%) | Fat (%) | Protein (%) | Carbohydrate (%) | Trend |
|---|---|---|---|---|---|---|---|
| D1 | 54.62 ± 0.81[c] (53.58 − 56.22) | 1.41 ± 0.05[e] (1.33 − 1.41) | 4.11 ± 0.03[c] (4.06 − 4.15) | 7.23 ± 0.11[b] (7.00 − 7.36) | 2.04 ± 0.01[c] (2.02 − 2.06) | 30.60 ± 0.87[a] (28.86 − 31.61) | M>C>F>Fi>P>A |
| D2 | 55.29 ± 1.52[c] (52.86 − 58.08) | 1.64 ± 0.16[de] (1.32 − 1.81) | 4.04 ± 0.04[cd] (3.99 − 4.13) | 7.63 ± 0.27[b] (7.14 − 8.06) | 2.03 ± 0.01[c] (2.01 − 2.05) | 29.37 ± 1.70[a] (26.03 − 31.59) | M>C>F>Fi>P>A |
| D3 | 57.14 ± 1.50[bc] (55.23 − 60.10) | 1.74 ± 0.04[cd] (1.66 − 1.81) | 4.69 ± 0.01[b] (4.68 − 4.71) | 6.13 ± 0.07[cd] (5.99 − 6.24) | 3.09 ± 0.02[b] (3.06 − 3.13) | 27.21 ± 1.52[ab] (24.17 − 28.94) | M>C>F>Fi>P>A |
| D4 | 58.70 ± 2.15[bc] (54.91 − 62.35) | 1.35 ± 0.02[e] (1.30 − 1.39) | 3.61 ± 0.03[e] (3.56 − 3.64) | 6.52 ± 0.16[c] (6.20 − 6.69) | 2.80 ± 0.01[b] (2.78 − 2.82) | 27.03 ± 2.30[ab] (20.18 − 21.42) | M>C>F>Fi>P>A |
| D5 | 46.90 ± 0.52[d] (46.01 − 47.80) | 1.77 ± 0.06[cd] (1.67 − 1.76) | 12.30 ± 0.14[a] (12.06 − 12.55) | 14.50 ± 0.16[a] (14.28 − 14.81) | 3.55 ± 0.06[a] (3.45 − 3.65) | 20.97 ± 0.40[bc] (20.18 − 21.42) | M>C>F>Fi>P>A |
| D6 | 58.56 ± 0.16[bc] (58.35 − 58.87) | 1.52 ± 0.02[de] (1.49 − 1.56) | 3.75 ± 0.02[e] (3.71 − 3.79) | 2.52 ± 0.03[f] (2.47 − 2.58) | 3.12 ± 0.02[b] (3.10 − 3.15) | 30.53 ± 0.19[a] (30.18 − 30.84) | M>C>Fi>P>F>A |
| D7 | 69.16 ± 1.02[a] (67.15 − 70.46) | 2.04 ± 0.11[bc] (1.84 − 2.22) | 3.85 ± 0.05[cde] (3.75 − 3.91) | 5.67 ± 0.09[d] (5.53 − 5.83) | 3.54 ± 0.17[a] (3.31 − 3.86) | 15.75 ± 0.80[c] (14.46 − 17.22) | M>C>F>Fi>P>A |
| D8 | 64.19 ± 1.20[ab] (61.83 − 65.74) | 2.13 ± 0.12[b] (1.93 − 2.33) | 3.80 ± 0.03[de] (3.74 − 3.84) | 5.53 ± 0.11[d] (5.31 − 5.66) | 3.71 ± 0.11[a] (3.56 − 3.93) | 20.65 ± 1.09[bc] (19.25 − 22.81) | M>C>F>Fi>P>A |
| D9 | 60.77 ± 2.59[bc] (55.60 − 63.49) | 2.66 ± 0.05[a] (2.58 − 2.76) | 3.88 ± 0.02[cde] (3.84 − 3.92) | 3.38 ± 0.06[e] (3.28 − 3.48) | 3.67 ± 0.05[a] (3.58 − 3.73) | 25.64 ± 2.68[ab] (22.84 − 31.00) | M>C>Fi>P>F>A |

**Notes:**

Data presented are means ± standard error from three replications (*n* = 3) and values in bracket are the range. Different superscript alphabets in the same column indicate differences at *p* < 0.05 (ANOVA, Tukey's test).

* Wet weight basis. Abbreviation: GE, genotype.

($p < 0.05$). The moisture content in fruit is a critical physicochemical aspect that influences fruit taste, texture, appearance, and lifespan. The major component in the durian was moisture, ranging between 46.90 ± 0.52 to 69.16 ± 1.02%. This indicates that these wild edible *Durio* genotypes provide a good amount of water in food. The *D. oxleyanus* (D7) had a significantly high moisture content (69.16 ± 1.02%), while red-fleshed *D. graveolens* (D5) had the lowest moisture content (46.90 ± 0.52%). *Maninang et al. (2011)* reported that due to the high water content in durian pulp, the fresh Chanee durian cultivar had a shelf life of 3 days under ambient conditions, whereas *Belgis et al. (2016)* reported that lai (*D. kutejensis*) had moisture varying between 49.05 to 59.95%, and exhibited a longer shelf life of up to 5 days. The *D. dulcis*, *D. graveolens*, and *D. kutejensis* genotypes identified in these studies were found to have lifespans comparable to those reported in previous research by *Belgis et al. (2016)*.

The average ash content of the wild edible *Durio* genotypes ranged from 1.35 ± 0.02 to 2.66 ± 0.05%, which is lower than the average range of ash content of *D. zibethinus* varieties from Thailand reported by *Charoenkiatkul, Thiyajai & Judprasong (2016)* (2.9–4.3%) but similar to that of the Malaysian durian varieties reported by *Isa et al. (2019)* (1.15–3.50%). Different fruit varieties with different localities can result in differences in the physicochemical properties of fruit. Previous studies on the ash content of *D. dulcis* (1.3%) (*Susi, 2017*), *D. graveolens* (yellow-fleshed) (1.11%) (*Nasaruddin, Noor & Mamat, 2013*), and *D. oxleyanus* (2.2%) (*Suwardi et al., 2022*) have reported similar results. The quantity of ash provides insight into the overall presence of minerals within it (*Kalsum & Mirfat, 2014*).

There was a significant difference among all of the genotypes regarding fibre content, which ranged from 3.61 ± 0.03 to 12.30 ± 0.14%, which is similar to the range of *D. zibethinus in* Malaysia cultivars (2.81–3.19%) (*Isa et al., 2019*), as determined by excluding the red-fleshed *D. graveolens* (D5). The fibre content of the red-fleshed *D. graveolens* (D5) was triple times higher (12.30 ± 0.14%) than that of the other genotypes. The average fibre content of the *D. dulcis* (D1 and D2) genotypes, on average, ranged from 4.04 ± 0.04 to 4.11 ± 0.03%, which was lower than that reported by *Susi (2017)* (10.19%). The fibre contents of the yellow-fleshed *D. graveolens* (D3) and *D. oxleyanus* (D7 and D8) genotypes were similar to those previously reported by *Nasaruddin, Noor & Mamat (2013)* (3.66%) and *Suwardi et al. (2022)* (3.5%), respectively. The recommended daily dietary fibre intake is 28 g/day for adult women and 36 g/day for adult men. Consuming 100 g of red-fleshed *D. graveolens* (D5) can fulfil approximately 34.17 to 43.94% of our body's daily fibre content. The high fibre content of durian, including both soluble and insoluble fibre, contributes to various health benefits, including improved cardiovascular health, weight management, and gastrointestinal function (*Slavin, 2013*).

Compared with those of the other wild edible *Durio* genotypes, the red-fleshed *D. graveolens* (D5) had significantly (two times) greater levels of fat (14.50 ± 0.16%), which resulted in its creamy taste pulp. The fat content of the other eight wild edible *Durio* genotypes ranged from 2.52 ± 0.03 to 7.63 ± 0.27%, which higher compared to the fat content of *D. zibethinus* cultivars, which ranged from 1.09-2.43% (*Isa et al., 2019*). The fat contents of *D. dulcis* genotypes (D1 and D2), yellow-fleshed *D. graveolens* (D3), and

*D. kutejensis* (D6) were similar to those in previous reports (*Nasaruddin, Noor & Mamat, 2013*; *Sunaryo et al., 2015*; *Susi, 2017*). Fat yields more than twice the energy per unit mass compared to carbohydrate and protein, making durian an energy-dense tropical fruit (*Kalsum & Mirfat, 2014*). Fat contributes to the development and release of aromatic compounds, influencing the overall flavour profile and sensory appeal of the fruit (*Sujang et al., 2023*).

The protein content of the wild edible *Durio* genotypes ranged from 2.03 ± 0.01 to 3.71 ± 0.11% within the range of the other wild edible *Durio* genotypes previously studied, namely, yellow-fleshed *D. graveolens* (3.08%) (*Nasaruddin, Noor & Mamat, 2013*), *D. kutejensis* cultivars (1.94–2.83%) (*Belgis et al., 2016*), *D. oxleyanus* (3.3%) (*Suwardi et al., 2022*) and *D. zibethinus* varieties (0.65–2.62%) (*Isa et al., 2019*). *Durio dulcis* genotypes (D1 and D2) were found to have lower protein content (2.03 ± 0.01 to 2.04 ± 0.01%); however, as proteins are composed of amino acids, *Susi (2017)* highlighted that *D. dulcis* provides essential amino acids such as tyrosine and valine, which are needed for tissue repair and various metabolic functions. The unique scent of durian arises from the synthesis of aromatic compounds by specific proteins during enzymatic reactions (*Li et al., 2018*; *Voon et al., 2007*; *Sujang et al., 2023*; *Peng, 2019*). The carbohydrate content of the wild edible *Durio* genotypes ranged from 15.75 ± 0.80% to 30.60 ± 0.87%, which is comparable to the range reported by *Nasaruddin, Noor & Mamat (2013)* (yellow-fleshed of *D. graveolens*, 20.18%) and *Isa et al. (2019)* (*D. zibethinus* cultivars, 27.66–35.44%). The carbohydrate contents of *D. dulcis* (D1 and D2) and *D. kutejensis* (D6) were two times greater than that of *D. oxleyanus* (D7). According to *Belgis et al. (2016)*, the carbohydrate content is related to the sugar content and sweet taste, which, in accordance with the total sugar of *D. dulcis* genotypes, was the second sweetest among the wild edible *Durio* genotypes. The relatively lower concentration of carbohydrates in some fruits is due to their high water content, a trend which can be seen in *D. oxleyanus* (D7 and D8) genotypes.

## Mineral composition of wild edible *Durio* genotypes

Minerals are essential nutrients found in fruit that play a vital role in human health. Compared with Thai *D. zibethinus* varieties (2.9–4.3%), wild edible *Durio* genotypes have a lower ash content (1.35–2.66%), but they are within the range of Malaysian varieties (1.15–3.50%). However, they still provide adequate essential minerals for health benefits. Table 5 presents the macronutrient and micronutrient composition, with the recommended dietary allowance (RDA) percentage for adults aged 19 to 50 years, of the wild edible *Durio* genotypes. The recommended dietary allowance (RDA) is a scientifically determined nutrient intake level established by the Food and Nutrition Board to meet the needs of nearly all healthy individuals. The present study revealed that different genotypes of wild edible *Durio* fruits had varied mineral concentrations, even within similar species, likely due to environmental factors influencing their mineral compositions. In terms of macronutrients, potassium (K) had the highest concentration, followed by phosphorus (P), magnesium (Mg), sodium (Na) and calcium (Ca); a similar trend was observed for commercialized durian varieties.

**Table 5 Macronutrients and micronutrients of wild edible durian (serving size = per 100 g) and percentage (%) RDA for ♂–males and ♀–females, ages ranging from 19 to 50 years old.**

**Macronutrients**

| Genotypes | Potassium, K | | Phosphorus, P | | Calcium, Ca | | Sodium, Na | | Magnesium, Mg | | |
|---|---|---|---|---|---|---|---|---|---|---|---|
| | mg 100 g$^{-1}$ | % Adult | mg 100 g$^{-1}$ | % Adult | mg 100 g$^{-1}$ | % Adult | mg 100 g$^{-1}$ | % Adult | mg 100 g$^{-1}$ | % Adult | |
| | | ♂♀ | | ♂♀ | | ♂♀ | | ♂♀ | | ♂ | ♀ |
| D1 | 901.83$^{e}$ | 19.19 | 502.05$^{b}$ | 71.72 | 10.73$^{f}$ | 1.07 | 31.82$^{b}$ | 3.18 | 71.17$^{bc}$ | 17.79 | 22.96 |
| D2 | 785.67$^{f}$ | 16.72 | 554.50$^{ab}$ | 79.21 | 9.07$^{g}$ | 0.91 | 28.19$^{bc}$ | 2.82 | 79.33$^{a}$ | 19.83 | 25.59 |
| D3 | 1,259.50$^{a}$ | 26.80 | 540.63$^{ab}$ | 77.23 | 18.82$^{a}$ | 1.88 | 49.90$^{a}$ | 4.99 | 78.67$^{ab}$ | 19.67 | 25.38 |
| D4 | 794.00$^{f}$ | 16.89 | 603.59$^{ab}$ | 86.23 | 11.20$^{e}$ | 1.12 | 26.88$^{bc}$ | 2.69 | 55.00$^{d}$ | 13.75 | 17.74 |
| D5 | 1,010.67$^{c}$ | 21.50 | 605.63$^{a}$ | 86.52 | 6.31$^{h}$ | 0.63 | 25.82$^{bc}$ | 2.58 | 39.67$^{e}$ | 9.92 | 12.80 |
| D6 | 966.67$^{d}$ | 20.57 | 535.17$^{ab}$ | 76.45 | 18.18$^{b}$ | 1.82 | 18.34$^{de}$ | 1.83 | 84.67$^{a}$ | 21.17 | 27.31 |
| D7 | 738.17$^{g}$ | 15.71 | 548.09$^{ab}$ | 78.30 | 12.29$^{d}$ | 1.23 | 24.93$^{cd}$ | 2.49 | 37.33$^{e}$ | 9.33 | 12.04 |
| D8 | 1,112.00$^{b}$ | 2.37 | 507.76$^{ab}$ | 72.54 | 14.96$^{c}$ | 1.50 | 25.02$^{cd}$ | 2.50 | 66.33$^{c}$ | 16.58 | 21.40 |
| D9 | 1,256.83$^{a}$ | 2.67 | 524.00$^{ab}$ | 74.86 | 6.36$^{h}$ | 0.64 | 14.61$^{e}$ | 1.46 | 70.33$^{c}$ | 17.58 | 22.69 |

**Micronutrients**

| Genotypes | Copper, Cu | | Iron, Fe | | | Zinc, Zn | | | Manganese, Mn | | |
|---|---|---|---|---|---|---|---|---|---|---|---|
| | mg 100 g$^{-1}$ | % Adult | mg 100 g$^{-1}$ | % Adult | | mg 100 g$^{-1}$ | % Adult | | mg 100 g$^{-1}$ | % Adult | |
| | | ♂♀ | | ♂ | ♀ | | ♂ | ♀ | | ♂ | ♀ |
| D1 | 0.13$^{cd}$ | 14.44 | 2.25 | 28.13 | 12.50 | 1.36$^{b}$ | 12.35 | 16.98 | 0.71$^{c}$ | 30.72 | 39.26 |
| D2 | 0.11$^{ef}$ | 12.22 | 2.84 | 35.54 | 15.80 | 1.44$^{b}$ | 13.05 | 17.94 | 0.51$^{de}$ | 22.32 | 28.52 |
| D3 | 0.12$^{def}$ | 13.33 | 2.09 | 26.10 | 11.60 | 1.34$^{b}$ | 12.17 | 16.73 | 0.41$^{fg}$ | 17.75 | 22.69 |
| D4 | 0.11$^{f}$ | 12.22 | 2.52 | 31.54 | 14.02 | 1.50$^{b}$ | 13.59 | 18.69 | 0.57$^{d}$ | 24.78 | 31.67 |
| D5 | 0.15$^{b}$ | 16.67 | 2.33 | 29.06 | 12.92 | 1.34$^{b}$ | 12.14 | 16.69 | 0.34$^{g}$ | 14.71 | 18.80 |
| D6 | 0.07$^{g}$ | 7.78 | 2.93 | 36.63 | 16.28 | 1.93$^{a}$ | 17.56 | 24.15 | 0.48$^{ef}$ | 21.01 | 26.85 |
| D7 | 0.13$^{c}$ | 14.44 | 3.90 | 48.73 | 21.66 | 0.80$^{d}$ | 7.27 | 10.00 | 0.77$^{bc}$ | 33.41 | 42.69 |
| D8 | 0.16$^{a}$ | 17.78 | 4.21 | 52.65 | 23.40 | 1.11$^{c}$ | 10.11 | 13.90 | 0.84$^{b}$ | 36.52 | 46.67 |
| D9 | 0.12$^{de}$ | 13.33 | 1.80 | 22.52 | 10.01 | 0.89$^{d}$ | 8.11 | 11.15 | 0.99$^{a}$ | 42.83 | 54.72 |

**Note:**
Different superscript alphabets in the same column indicate differences at $p < 0.05$ (ANOVA, Tukey's test). Values are given as means (mg). ♂-male and ♀-female.

Potassium levels ranged from 738.17 mg 100 g$^{-1}$ for *D. oxleyanus* (D8) to 1,259.50 mg 100 g$^{-1}$ for yellow-fleshed *D. graveolens* (D9). Although the *D. graveolens* (yellow, orange, and red-fleshed) and *D. oxleyanus* (D7 and D8) genotypes are the same, this genotype has significantly different potassium properties. The potassium content of this wild edible durian was close to that of *D. zibethinus* varieties, ranging from 800 to 1,100 mg 100 g$^{-1}$ (*Haruenkit et al., 2007*; *Aziz & Jalil, 2019*). A total of 100 g of wild edible durian flesh provides 111.20 to 1,259.50 mg of potassium, or approximately 15.71 to 26.80% of the daily estimated recommendation of 4,700 mg. This shows that the K content in wild edible durians can reach approximately 15–27% of our body's daily requirements. Potassium is crucial for many important physiological functions, such as nerve and muscle activity, hormone balance, blood pressure regulation, digestion, and proper acid–base balance and

metabolism (*Haugen et al., 2018*). Notably, patients with end-stage renal failure should limit their intake of durians, as durians have a greater K content than bananas and jackfruit, which can cause hyperkalaemia (*Bansal & Pergola, 2020*). Phosphorus is important for the fundamental process of metabolism in the body and provides strength and rigidity to bones and teeth (*Wardlaw et al., 2003*). There was a significant difference between the lowest and highest amounts of P in *D. dulcis* (D1) ($502.05 \pm 8.99$ mg 100 g$^{-1}$) and red-fleshed *D. graveolens* (D5) ($605.63 \pm 6.46$ mg 100 g$^{-1}$), respectively. The results of the present study are approximately twenty times higher than those reported for wild edible durian by *Hoe & Siong (1999)* and *D. zibethinus* of Thailand varieties (*Charoenkiatkul, Thiyajai & Judprasong, 2016*; *Haruenkit et al., 2007*). The RDA for Na intake does not exceed than 700 mg per day. Phosphorus deficiency can impair bone health and quality of life, while excessive intake can be a risk factor for cardiovascular disease (*Takeda et al., 2012*). Therefore, it is advisable to consume these wild edible durians in moderate quantities, as they supply up to 71.72–86.52% of the daily P requirement.

In the present study, the amount of sodium present in wild edible durian ranged between $14.61 \pm 0.02$ mg 100 g$^{-1}$ and $49.90 \pm 2.77$ mg 100 g$^{-1}$. The highest amount of Na ($49.90 \pm 2.77$ mg 100 g$^{-1}$) was detected in yellow-fleshed *D. graveolens* (D3), and the lowest amount ($14.61 \pm 0.02$ mg 100 g$^{-1}$), and the lowest amount was in *D. zibethinus* (D9). The Na concentration in *D. zibethinus* of Thailand varieties reported by *Charoenkiatkul, Thiyajai & Judprasong (2016)*, is 37–67 mg 100 g$^{-1}$, which is similar to the range used in the present study. The amount of Na in wild edible durian is acceptable, as the recommended daily Na intake is 1,500 mg for both female and male adults (19–50 years old). According to *McDonough (2010)*, Na is important for regulating blood volume in order to prevent hypotension; however, excessive sodium intake is linked to hypertension and increased cardiovascular diseases.

Adequate intake of calcium and magnesium supports carbohydrate and protein metabolism, bone formation, electrolyte balance and tissue stability. Durian provides a large dose of Mg per serving, making it ideal for bone health. The Mg content in *D. kutejensis* (D6) was the highest, at $84.67 \pm 0.29$ mg 100 g$^{-1}$, which was two times greater than that in *D. oxleyanus* (D7), at $37.33 \pm 0.29$ mg 100 g$^{-1}$. A previous study by *Hoe & Siong (1999)* reported that yellow-fleshed *D. graveolens* and *D. kutejensis* possessed Mg contents of 27 mg 100 g$^{-1}$ and 19 mg 100 g$^{-1}$, respectively, which were 2–3 times lower than the present values. However, the Mg content of wild edible durian varieties of Thailand is within the range of *D. zibethinus* (*Charoenkiatkul, Thiyajai & Judprasong, 2016*; *Haruenkit et al., 2007*). The RDA intake of Mg is 400–420 mg for adult males and 310–320 mg for adult females. Therefore, 100 g of the flesh of the wild edible durian is able to provide up to 9.92–21.17% and 12.04–27.31% to both male and female adults, respectively.

Calcium is a key macronutrient in fruit that plays a role in health benefits such as maintaining bone health and promoting youth bone mass (*Nieves, 2009*). The Ca levels in wild edible durians varied between $6.31 \pm 0.02$ to $18.82 \pm 0.18$ mg 100$^{-1}$ within the values reported for *D. zibethinus* from Thailand by *Charoenkiatkul, Thiyajai & Judprasong (2016)* (9–17 mg 100 g$^{-1}$) and *Haruenkit et al. (2007)* (18.1 mg 100 g$^{-1}$). Among these, the highest

and lowest Mg contents were found in yellow-fleshed and red-fleshed *D. graveolens*, respectively. The Ca content of yellow-fleshed *D. graveolens* in a previous report (*Hoe & Siong, 1999*) was slightly lower than that in the present study. A small amount of Ca supplies 0.63 to 1.88% of the daily Ca requirement for adult males and females.

Wild edible durian contains small amounts of trace minerals such as copper (Cu), iron (Fe), zinc (Zn) and manganese (Mn). The trend of micronutrients in wild edible durian was Mn > Cu > Zn > Fe except for D5, D7, and D8, with a trend of Mn > Zn > Cu > Fe and D9, where zinc was the most abundant. Among the studied species, the Fe content of *D. oxleyanus* was significantly greater, at $3.90 \pm 0.27$ mg 100 $g^{-1}$ and $4.21 \pm 0.04$ mg 100 $g^{-1}$, respectively, and the lowest content was recorded in *D. zibethinus* (D9), at $1.80 \pm 0.01$ mg 100 $g^{-1}$. The RDA for Fe listed by gender and age between 19–50 years old, 8 mg per day for adult males and 18 mg per day for adult females, appears sufficient in most individuals. Two servings (100 g) of *D. oxleyanus* meet the daily requirement of Fe. The human body requires a sufficient amount of Fe every day to prevent anaemia, because Fe aids in the formation of the haemoglobin in red blood cells that carries oxygen from the lungs to all cells of the body. The wild edible durians boast a Cu content of less than 0.20 mg per 100 g serving, providing between 7.78% (*D. kutejensis*, D6) and 17.78% (D8) of the daily Cu requirement for adults, adding to their nutritional appeal. Per serving (100 g) of wild edible durian can provide up to 7.27% in *D. oxleyanus* (D7) to 24.15% in *D. kutejensis* (D6) and 14.71% in red-fleshed *D. graveolens* to 54.72% in *D. zibethinus* (D9) of the RDA for Zn and Mn, respectively. Despite their small quantities, these microminerals serve as nutritional supplements and contribute to dietary diversification and optimize micronutrient levels for human health, such as immune function, energy metabolism, bone health, and antioxidant defence (*Nieder et al., 2018*).

## Phytochemical properties of wild edible *Durio* genotypes

Generally, recognized as a nutritious fruit, durians are abundant in antioxidants and various bioactive compounds known for their potential positive effects on health. *Yee (2021)* specifically documented the bioactive compounds and antioxidant activity in *Durio zibethinus* cultivars. Nonetheless, there is limited literature elucidating the bioactive and antioxidant characteristics of wild edible *Durio* species. It is thought that vitamin C is a potent antioxidant in fruit and vegetables. The amount of ascorbic acid varied significantly ($p < 0.05$) among the wild edible *Durio* genotypes (Table 6). The concentration of ascorbic acid was found to be the highest in *D. kutejensis* (D6), at $61.56 \pm 4.83$ mg 100 $g^{-1}$, and the lowest at $10.47 \pm 0.20$ mg 100 $g^{-1}$ in *D. zibethinus* (D9). This is in line with *D. kutejensis* having a bright flesh colour, which is yellow-orange to orange, indicating the abundance of carotene as well as pro-vitamin A (*Krismawati, 2012*). This ascorbic acid range in wild edible durian genotypes is similar to the range ($24.71–63.29$ mg 100 $g^{-1}$) of *D. zibethinus* cultivars from Malaysia reported by *Isa et al. (2019)*. In contrast, the present values for *D. dulcis* genotypes (D1 and D2) were two times higher than the reported amount ($27.45$ mg 100 $g^{-1}$) (*Susi, 2017*). This titration of ascorbic acid can be inaccurate due to limitations in endpoint detection, difficulty in analysing coloured extracts, and inability to accurately determine endpoints. This suggests that these wild edible *Durio* species can be a

**Table 6 The bioactive and antioxidant properties of wild edible *Durio* genotypes.**

| Genotypes | | AA (mg 100 g⁻¹) | CAR (µg 100 g⁻¹ FW) | TPC (mg GAE 100 g⁻¹) | TFC (mg QE 100 g⁻¹) | DPPH (mg mL⁻¹) | FRAP (µM g⁻¹) |
|---|---|---|---|---|---|---|---|
| D1 | *D. dulcis* | 45.37 ± 1.75[b] (41.88 – 47.12) | 402.85 ± 7.55[e] (391.32 – 417.05) | 72.48 ± 2.66[a] (67.17 – 75.29) | 13.81 ± 2.62[a] (10.07 – 18.87) | 8.28 ± 0.39[ab] (7.52 – 8.78) | 1.68 ± 0.02[bcd] (1.64 – 1.71) |
| D2 | *D. dulcis* | 45.99 ± 0.75[b] (46.20 – 47.15) | 384.45 ± 3.44[e] (378.25 – 390.10) | 67.95 ± 0.97[ab] (66.88 – 69.88) | 8.71 ± 1.02[ab] (7.30 – 10.68) | 9.38 ± 0.21[ab] (8.97 – 9.64) | 1.37 ± 0.05[de] (1.28 – 1.45) |
| D3 | *D. graveolens* (yellow-fleshed) | 31.41 ± 0.00[c] (31.41) | 976.36 ± 3.99[d] (968.44 – 981.17) | 44.43 ± 2.14[d] (40.15 – 46.58) | 6.63 ± 2.34[b] (2.47 – 10.55) | 10.42 ± 0.42[a] (9.70 – 11.17) | 1.37 ± 0.02[de] (1.33 – 1.39) |
| D4 | *D. graveolens* (orangee-fleshed) | 38.39 ± 1.74[bc] (36.65 – 41.88) | 1,646.18 ± 4.52[b] (1,640.38 – 1,655.09) | 29.53 ± 1.09[e] (27.85 – 31.58) | 6.07 ± 0.68[b] (5.14 – 7.39) | 6.93 ± 0.29[bc] (6.63 – 7.52) | 1.34 ± 0.01[e] (1.32 – 1.35) |
| D5 | *D. graveolens* (red-fleshed) | 40.14 ± 4.62[bc] (31.41 – 47.12) | 2,627.18 ± 6.03[a] (2,615.12 – 2,633.20) | 47.43 ± 4.48[cd] (42.05 – 56.32) | 11.24 ± 0.53[ab] (10.19 – 11.83) | 6.57 ± 0.70[bc] (5.17 – 7.37) | 1.46 ± 0.01[cde] (1.44 – 1.48) |
| D6 | *D. kutejensis* | 61.56 ± 4.83[a] (52.47 – 68.91) | 1,022.81 ± 9.25[c] (1,011.57 – 1,041.15) | 67.02 ± 2.46[ab] (62.83 – 71.36) | 9.15 ± 0.63[ab] (8.21 – 10.35) | 5.03 ± 0.30[c] (4.43 – 5.34) | 1.92 ± 0.10[ab] (1.75 – 2.10) |
| D7 | *D. oxleyanus* | 13.41 ± 1.52[d] (10.54 – 15.72) | 294.97 ± 3.70[g] (290.56 – 302.33) | 74.77 ± 1.50[a] (72.71 – 77.69) | 9.57 ± 0.19[ab] (9.33 – 9.94) | 8.59 ± 0.43[ab] (8.07 – 9.43) | 1.96 ± 0.07[ab] (1.89 – 2.09) |
| D8 | *D. oxleyanus* | 13.96 ± 1.75[d] (10.47 – 15.71) | 328.54 ± 5.89[f] (322.69 – 336.68) | 57.18 ± 1.37[bc] (54.54 – 59.11) | 8.92 ± 0.36[ab] 8.26 – 9.52) | 9.34 ± 0.35[ab] (8.68 – 9.84) | 2.25 ± 0.14[a] (2.02 – 2.49) |
| D9 | *D. zibethinus* | 10.47 ± 0.20[d] (10.10 – 10.77) | 331.72 ± 4.52[f] (322.69 – 336.68) | 48.37 ± 1.52[cd] (45.67 – 50.93) | 6.14 ± 0.10[b] (6.00 – 6.33) | 7.40 ± 1.53[abc] (4.35 – 9.09) | 1.42 ± 0.06[cde] (1.37 – 1.54) |

**Note:**
Data presented are means ± standard error (SE) from three replications ($n = 3$). Different superscript alphabets in the same column indicate differences at $p < 0.05$ (ANOVA, Tukey's test). Abbreviations: AA, ascorbic acid; CAR, carotenoid; TPC, total phenolic content; TFC, total flavonoid content; DPPH, 2,2-diphenyl-1-picryl-hydrazl-hydrate radical scavenging activity ; FRAP, ferric reducing antioxidant power assay.

source of vitamin C for rural communities in Borneo, as these wild edible *Durio* species are commonly found in primary or secondary forests (*Sujang et al., 2023*). *Leong & Shui (2002)* reported that individuals consuming consistently higher levels of dietary ascorbic acid have a reduced likelihood of developing cancer. Moreover, vitamin C catalyses vital physiological processes and acts as an antioxidant and has higher antioxidant potency than vitamins A and E (*Suwardi et al., 2022*). The RDA for vitamin C is 75 mg for women and 90 mg for men aged between 19 to 70 years old, based on its role as an antioxidant and protection from deficiency. A serving of 100 g of *D. kutejensis* meets approximately 81.33% and 67.78% of the daily requirement for vitamin C for female and male adults, respectively.

Carotenoids are a group of pigments responsible for the appearance and attractiveness of fruit that influences the colouration of durian flesh and can be used as markers to distinguish between durian clones and assess fruit quality (*Wisutiamonkul et al., 2017*). The wild edible *Durio* fruits in these studies had aesthetically coloured pulp ranging from yellow and orange to red, which is assumed to reflect a high total carotenoid content. As expected, red-fleshed *D. graveolens* (D5), which has a bright red-fleshed colour, had a significantly four times higher in total carotenoid content (2,627.18 µg 100 g⁻¹) than did *D. oxleyanus* (D7), which has a pale yellow-fleshed colour and the lowest carotenoid content (294.97 µg 100 g⁻¹). The bright red flesh of *D. graveolens* is associated with a high total carotenoid content, probably due to the presence of lycopene, which absorbs more of the visible spectrum because of its many conjugated carbon double bonds that accumulate

in large amounts in red-fleshed *D. graveolens*. Carotenoids are an important product quality attribute, making them visually attractive and becoming a major breeding trait.

Polyphenols, such as flavonoids and phenolic acids, are highly active compounds found abundantly in durian (*Yee, 2021*). There were significant differences in the polyphenol contents of the wild edible *Durio* fruits, which ranged from 29.53 to 74.77 mg gallic acid (GAE) 100 g$^{-1}$ in total phenolic content (TPC) and from 6.07 to 13.81 mg quercetin (QE) 100 g$^{-1}$ in total flavonoid content (TFC). The range of phenolic and flavonoid contents of the wild edible durian was in accordance with the range of common durian cultivars (Chaer Poy, Ang Jin, D11, and Yah Kang) reported by *Ashraf, Maah & Yusoff (2010)*. The highest total phenolic content was found in *D. oxleyanus* (D7) and *D. dulcis* (D1), at 74.77 ± 1.50 and 72.48 ± 2.66 mg GAE 100 g$^{-1}$, respectively, and orange-fleshed *D. graveolens* (D4) exhibited two times lower phenolic content, at 29.53 ± 1.09 mg GAE 100 g$^{-1}$. The high TPC in *D. dulcis* might be due to the high concentration of ascorbic acid. The TPC of *D. dulcis* (D1 and D2) was similar to the TPC of *D. dulcis* reported by *Ikram et al. (2009)* (183.06 mg 100 g$^{-1}$).

Wild edible durian with bright-coloured flesh, such as *D. kutejensis* (D6) with orange to yellow-coloured flesh, red-fleshed *D. graveolens* (D5), and orange-fleshed *D. graveolens* (D4), exhibited lower EC$_{50}$ values of 5.03 ± 0.30 mg mL$^{-1}$, 6.57 ± 0.70 mg mL$^{-1}$, and 6.93 ± 0.29 mg mL$^{-1}$, respectively. A lower EC$_{50}$ value indicates strong antioxidant properties. Antioxidants able to reduce Fe$^{3+}$ to Fe$^{2+}$ were most active in *D. oxleyanus* (D7 and D8) genotypes (1.96 and 2.25 µM g$^{-1}$) and *D. kutejensis* (D6) (1.92 µM g$^{-1}$), where significantly high FRAP values were observed, indicating high antioxidant capacity. According to the DPPH and FRAP assays, *Durio kutejensis* (D6) had the highest antioxidant capacity among the wild edible Durio genotypes, showing a trend towards high levels of vitamin C and polyphenolic compounds. The higher antioxidant activity could also be related to the range and red flesh with a higher total carotenoid content. This trend has been reported by *Isa et al. (2019)*, who showed that the orange flesh of Udang Merah fruits exhibited higher phenolic and carotenoid contents with increased antioxidant activity.

## Sensory analysis of the wild edible *Durio* species

The sensory attributes of foods include their appearance, odour or aroma, texture, and flavour. The sensory attributes of the flesh of wild durian fruits are based on their appearance (brightness and dullness), flavour (sweetness and bitterness), taste (creaminess, smoothness, juiciness, and stickiness), and overall taste. The sensory evaluation illustrated in a spider web, as shown in Fig. 2, was evaluated by 15 trained panellists using quantitative descriptive analysis (QDA). The profiling of volatile organic compounds and the aroma attributes of these wild edible durians that are present in Sarawak was performed in our previous study (*Sujang et al., 2023*). Sensory evaluation is important for assessing different fruit clones for fresh consumption, including colour, texture, flavour, and aftertaste. The appearance of intense and bright colours (red, orange, and yellow) from *D. graveolens* and *D. kutejensis* attracted the panellists more, as both species had flesh colour that was bright, while *D. zibethinus* and *D. oxleyanus* species were categorized as having dull colours because they had the least intense colour. According to a

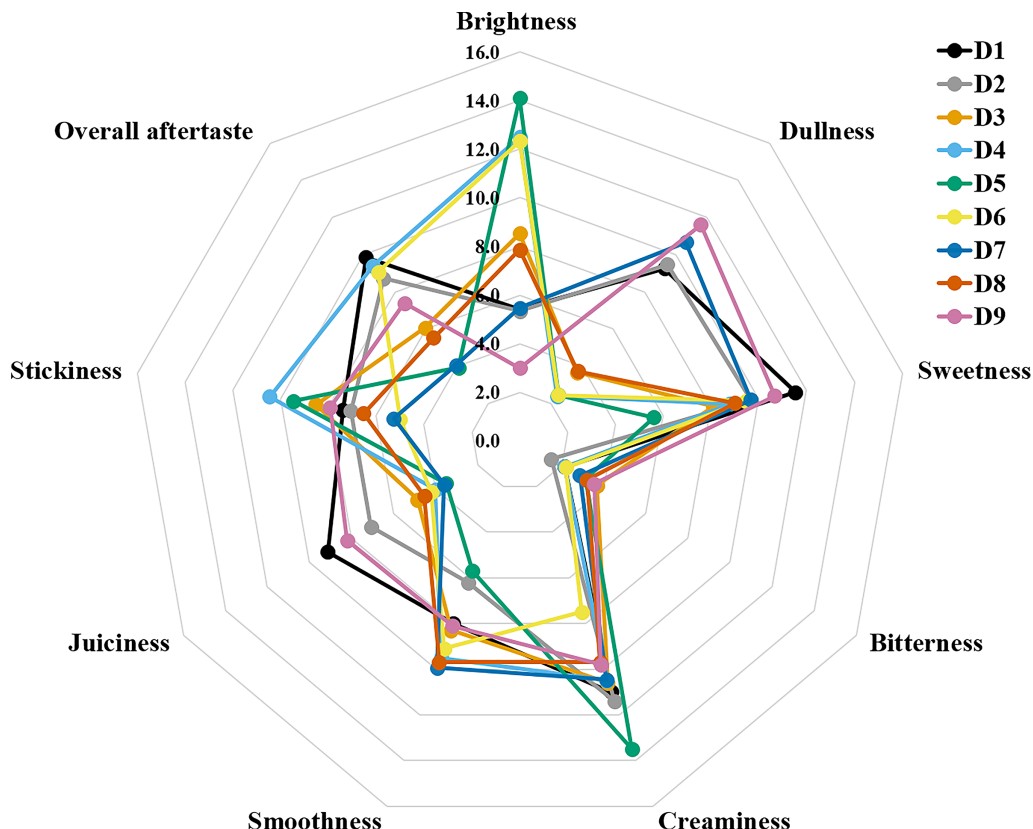

**Figure 2** The spider web of nine sensory attributes in nine edible wild durians that are present in Sarawak.

previous study of wild edible durian (*Sujang et al., 2023*), not only is the flesh of wild edible durian appealing due to its colouration, but also its exocarp. This aesthetic value should increase its consumption rate (*Susilawati & Sabran, 2018*).

The primary factors influencing consumer preference for fruits and vegetables are their flavour and aroma. Essential tastes such as sweetness is attributed to soluble sugars, sourness to organic acids, and bitterness is often derived from compounds such as phenolics, triterpenes, or certain aldehydes (*Sánchez-Rodríguez et al., 2019*). Wild edible durians perceived a moderate score for sweet taste except for red-fleshed *D. graveolens*, whose sugar concentration is the lowest among the genotypes. Orange-flesh plants of *D. graveolens*, *D. oxleyanus* and *D. zibethinus* were slightly bitter, which is probably related to the occurrence and content of polyphenols. The bitter taste of durian is likely linked to specific amino acids, including alanine, proline, phenylalanine, and isoleucine, which have been shown to contribute to the perception of bitterness (*Xiao, Niu & Niu, 2022*). Concurrently with the twofold fat content among the genotypes, red-fleshed *D. graveolens* was also perceived as the creamiest and stickiest by the panellists, influenced by the low score of smooth texture and juiciness in the flesh. *Durio dulcis* has the second highest fat content and was also noted by the trained panellists as having the second creamiest flesh among the nine genotypes. *Durio kutejensis* and *D. oxleyanus* had the smoothest mouthfeel

among the nine genotypes and had the least sticky flesh, except for *D. graveolens* (orange-fleshed), which had the stickiest flesh among the genotypes, although it was perceived as smooth-texture flesh. *D. dulcis* was the most juicy among the genotypes, as half of its proximate constituent was moisture content, followed by *D. zibethinus*. Durian is known for its complex and distinctive flavour profile and unique combination of sweet, savoury, and sometimes pungent elements. *Durio dulcis*, *D. graveolens* (orange-fleshed), *D. kutejensis* and *D. zibethinus* (terung iban) had high scores for overall aftertaste. *Durio dulcis* might leave a lingering impression on the taste buds after consumption, as it has the worst smell among the *Durio* species. The overall aftertaste was least apparent in *D. graveolens* (red-fleshed), probably due to its bland taste when consumed raw. A high or low overall aftertaste score may indicate both positive (well-balanced and enjoyable, pleasant and enduring) and negative (bitterness, astringency, overpowering or off-putting element) qualities. Durian aftertaste, like its overall flavour, is subjective and varies among individuals based on personal preferences.

## Principal component analysis of the chemical compositions of the wild edible *Durio* species

Principal component analysis (PCA) was conducted to establish the relationships between the physicochemical composition, proximate composition and phytochemical properties (Fig. 3). The PCA explained 76.55% of the variance of the dataset in two dimensions, PC1 (42.87%) and PC2 (33.68%). The PCA results revealed that the nine genotypes of wild edible durian were classified into four groups. In the factor plot of both the positive side of PC1 and PC2 (Group 1), yellow-fleshed *D. graveolens* (D3) and *D. zibethinus* (D9) were grouped together and had higher concentrations of total sugar in correlation with the total soluble solid, sucrose and maltose compositions. The *D. oxleyanus* genotypes (D7 and D8) grouped together in Group 2, located at the right of the positive side of PC2. *D. oxleyanus* had a slight to neutral pH value and had the highest protein content compared to the other wild edible durian genotypes. In the FRAP antioxidant assay, it had the greatest ability to reduce $Fe^{3+}$ to $Fe^{2+,}$ indicating that it has a high antioxidant capacity.

Orange- and red-fleshed *D. graveolens* (Group 3) had negative PC1 and PC2 values and were characterized by carotenoid content, fat, fibre, ABTS scavenging assay, and flavonoid content (TFC) data. Among the wild edible durian genotypes, the orange- and red-fleshed *D. graveolens* have the most attractive flesh colour, which corresponds to its high concentration of carotenoids. The red-fleshed *D. graveolens* fibre and fat contents four times greater than those of the other wild edible durian genotypes. High presence of flavonoids in both genotypes indicates that their antioxidant properties contribute to overall health and disease prevention. Both *D. dulcis* genotypes (D1 and D2) and *D. kutejensis* (D6) grouped together in Group 4 on the negative right side of PC2 and were characterized by their ascorbic acid content, total titratable acidity (TTA), carbohydrate content, fructose content, and glucose content. Among the wild edible durian genotypes, *Durio dulcis* and *D. kutejensis* have high ascorbic acid and carbohydrate contents.

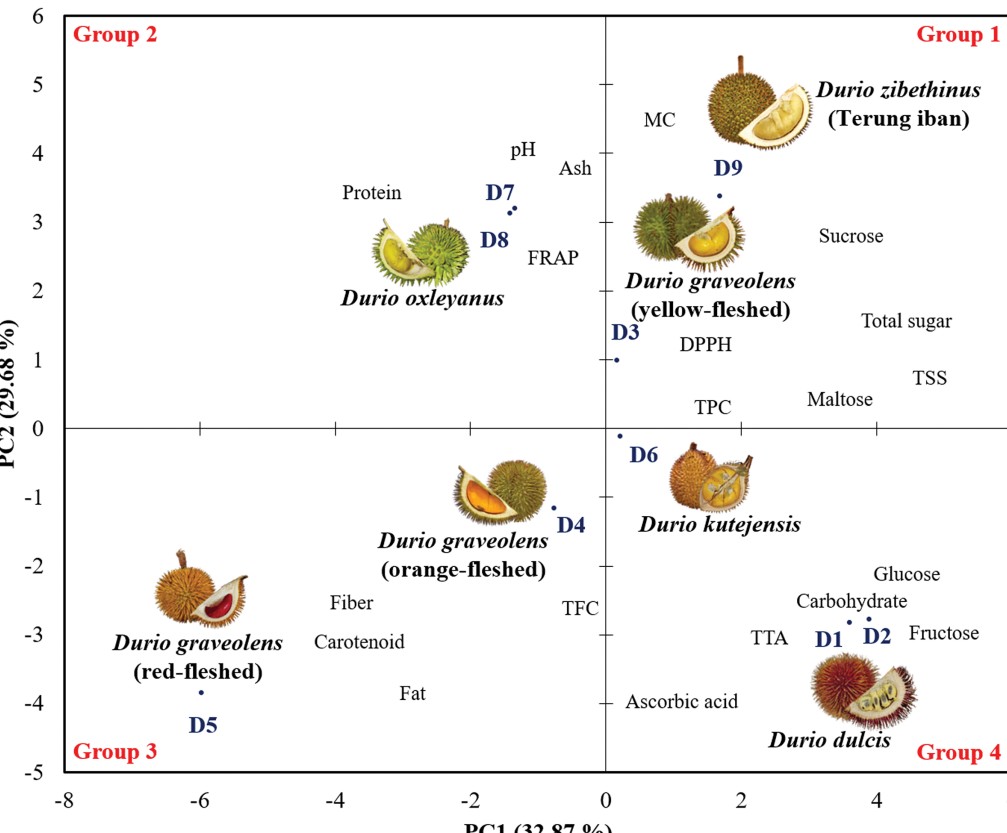

**Figure 3** Principal component analysis (PCA) biplot of wild edible durians in Sarawak.

**Table 7 Correlation coefficients between the sensory attributes and chemical properties of the nine genotypes of wild edible durians.**

| | Brightness | Dullness | Sweetness | Bitterness | Creaminess | Smoothness | Juiciness | Stickiness | Aftertaste |
|---|---|---|---|---|---|---|---|---|---|
| Moisture | −0.465 | 0.351 | 0.474 | 0.164 | −0.584 | **0.843** | −0.143 | −0.555 | −0.155 |
| Fiber | 0.534 | −0.348 | **−0.806** | 0.254 | **0.734** | **−0.678** | −0.320 | 0.415 | −0.533 |
| Fat | 0.557 | −0.375 | **−0.739** | −0.089 | **0.829** | **−0.673** | −0.309 | 0.431 | −0.363 |
| Protein | 0.154 | −0.185 | −0.405 | **0.792** | −0.132 | 0.318 | −0.532 | −0.085 | **−0.735** |
| Carb | −0.064 | −0.011 | 0.398 | −0.453 | −0.209 | −0.280 | 0.628 | 0.204 | **0.881** |
| pH | −0.113 | −0.143 | −0.096 | **0.679** | −0.339 | 0.628 | −0.418 | −0.291 | −0.545 |
| TSS | **−0.764** | **0.678** | **0.916** | −0.472 | −0.390 | 0.178 | **0.734** | −0.371 | 0.651 |
| Fructose | −0.535 | 0.549 | 0.575 | −0.571 | 0.221 | −0.432 | **0.840** | 0.022 | 0.606 |
| Glucose | −0.552 | 0.573 | 0.560 | −0.573 | 0.237 | −0.465 | **0.836** | −0.030 | 0.558 |
| Sucrose | −0.603 | 0.427 | **0.753** | 0.008 | −0.652 | 0.464 | 0.498 | −0.169 | 0.530 |
| Maltose | −0.501 | 0.425 | 0.507 | 0.018 | −0.328 | −0.226 | **0.708** | −0.073 | 0.504 |
| Total sugar | **−0.698** | 0.555 | **0.825** | −0.157 | −0.492 | 0.210 | **0.726** | −0.145 | 0.651 |

**Note:**
Significant correlations at the $p < 0.05$ level is marked in bold. Abbreviations: Carb, carbohydrate; TSS, total soluble sugar.

### Correlation analysis between chemical composition and sensory properties of the wild edible *Durio* species

Table 7 shows the correlation coefficients of the chemical composition and sensory attributes of the nine genotypes of wild edible durian. The results show that there is a highly positive correlation ($p < 0.05$) between moisture content and smoothness ($r = 0.843$), where juiciness in fruit is a texture attribute, as fruits with higher water content tend to be perceived as juicer when consumed. There was an inverse correlation between fibre and fat content and between sweetness ($r = -0.806$ and $r = -0.739$, respectively) and smoothness ($r = -0.678$ and $r = -0.673$, respectively), indicating that having high fibre and fat content does not directly correlate with a sweet taste and smooth texture of durian flesh. However, the presence of fibre and fat can enhance the perception of creaminess ($r = 0.734$ and $r = 0.829$, respectively) due to its ability to carry and amplify flavours, including sweetness, creating a creamy or cohesive texture, coating the taste buds and enhancing the release of flavour compounds. The pH of the wild edible durian was positively correlated with the pH and bitter taste ($r = 0.679$), where lower pH levels (more acidic) can intensify the perception of bitterness, and this is proven as the pH of the wild edible durian being slightly acidic. There was a correlation between the total soluble solids (TSS) content of fruit and the colour (brightness and dullness) of the fruit flesh, and there was a positive correlation between the TSS content and sweetness ($r = 0.916$). Typically, fruits with a relatively high TSS, which often correlates with increased sugar content, tend to have an intense colour in their flesh due to the accumulation of pigments such as carotenoids and anthocyanins. The sugar composition, fructose content, glucose content, and maltose content had positive correlations with juiciness ($r = 0.840$, $r = 0.836$, and $r = 0.708$, respectively); moreover, sucrose content had a positive correlation with sweetness ($r = 0.753$), as sucrose was the predominant sugar in the wild edible durian. A higher sugar content can result in a juicier mouthfeel due to the presence of dissolved sugars. A higher fat content contributes to creaminess, which can create a sensation of mouth coating or heaviness, which may reduce the perceived sweetness and juiciness by masking or diluting this sensation.

## CONCLUSIONS

This study revealed significant variations in chemical composition and sensory attributes among wild edible durians. Each of the genotypes has its own unique quality profile, contributing to health benefits depending upon its composition. *Durio dulcis* and *D. kutejensis*, which have relatively high total soluble solid (TSS) contents, exhibit sweeter tastes and more vibrant flesh colours. Sucrose, the predominant sugar in most wild edible durians contributes to their distinct sweet flavour profile, except for red-fleshed *D. graveolens*, where fructose is the dominant sugar, and it also stands out for its high fat content, which results in a creamy taste and texture. The unique chemical and sensory properties of various wild edible durians showcase their potential to attract consumers with diverse tastes and preferences. Potassium is abundant in wild edible durians, and these plants supply essential microminerals that support daily nutritional needs and enhance dietary diversity. *Durio graveolens* and *D. kutejensis*, which have attractive flesh

colours (yellow, orange, red), exhibit high levels of ascorbic acid and carotenoids, while the dull flesh of *D. dulcis* and *D. oxleyanus* contains high phenolic and flavonoid contents. Therefore, a moderate consumption approach is recommended to fully enjoy the distinct flavour of the indigenous *Durio* species. Overall, these findings provide valuable insights into the nutritional and sensory profiles of wild *Durio* species, offering opportunities for utilization and consumer preference in both the food and health industries and enhancing biodiversity conservation efforts in Sarawak.

## ACKNOWLEDGEMENTS

We extend our appreciation to Universiti Putra Malaysia for their assistance and facilities during the course of this study. Additionally, we extend our gratitude to the Sarawak Biodiversity Centre for granting approval under reference number JKM/SPU/608-8/2/1 Vol.3 to conduct research on indigenous durians from Sarawak. We express our sincere appreciation to the local communities, particularly in rural areas, for their exceptional hospitality and generous support us, willingly guiding us to explore the forest for sampling. Their kind guidance, provision of accommodation and assistance with transportation have been instrumental in the success of this research endeavour.

### Funding

This research was funded by Ministry of Higher Education Malaysia under the Fundamental Research Grant Scheme (FRGS), (FRGS/1/2020/STG03/UPM/02/10). The funders had no role in study design, data collection and analysis, decision to publish, or preparation of the manuscript.

### Grant Disclosures

The following grant information was disclosed by the authors:
Ministry of Higher Education Malaysia under the Fundamental Research Grant Scheme (FRGS): FRGS/1/2020/STG03/UPM/02/10.

### Competing Interests

The authors declare that they have no competing interests.

### Author Contributions

- Gerevieve Bangi Sujang conceived and designed the experiments, performed the experiments, analyzed the data, prepared figures and/or tables, authored or reviewed drafts of the article, and approved the final draft.
- Shiamala Devi Ramaiya conceived and designed the experiments, performed the experiments, analyzed the data, authored or reviewed drafts of the article, and approved the final draft.
- Shiou Yih Lee conceived and designed the experiments, analyzed the data, authored or reviewed drafts of the article, and approved the final draft.

- Muta Harah Zakaria analyzed the data, authored or reviewed drafts of the article, and approved the final draft.

## Field Study Permissions

The following information was supplied relating to field study approvals (*i.e.*, approving body and any reference numbers):

The Sarawak Biodiversity Centre for granted approval under reference number JKM/SPU/608-8/2/1 Vol.3.

## Data Availability

The raw measurements are available in the Supplemental Files.

## Supplemental Information

Supplemental information for this article can be found online at http://dx.doi.org/10.7717/peerj.17688#supplemental-information.

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
