# Peer review of "Characterization of indigenous Durio species from Sarawak, Borneo: relationships between chemical composition and sensory attributes"

_PeerJ, doi:10.7717/peerj.17688_

## Round 0.1 · original submission · Major Revisions

The reviewers indicate that extensive improvement of the grammar and style is needed. The size and sampling method are critical to further interpretation. A contextual comparison between the literature on commercial durian and wild varieties is also important for this paper.

**Language Note:** The review process has identified that the English language must be improved. PeerJ can provide language editing services - please contact us at [email protected] for pricing (be sure to provide your manuscript number and title). Alternatively, you should make your own arrangements to improve the language quality and provide details in your response letter. – PeerJ Staff

Reviewer 1 ·

Basic reporting

In the introduction section, durian is discussed in terms of the genus (Durio sp.) , which each region has various benefits. This does not include any species. But in reality This research is a study of wild durian, which is a different species from the introduction mentioned above. Therefore, the strength of this article should be strengthened by strengthening the focus on the physical properties of wild durian.

Experimental design

I would like to explain the ripe state of wild durians harvested . This is because in each stage of ripening the active substances and physical characteristics are different, which are caused by the enzymes inside the durian pulp.

Validity of the findings

Grouping with PCA is a good technique for showing the relationship between durian characters in each species.

Additional comments

The aroma volatile of durian is still another characteristic to be fingerprint displayed. Many articles have mentioned the unique characteristics of durian if they are separated according to their odor characteristics such as sulfur, esters, or even amino acids.

Reviewer 2 ·

Basic reporting

The document needs to be revised; there are some typos. Lines 104 and 105 ¿ What does RM mean? Line 138-142. Please describe this part of the methodology in depth: How was the pulp separated from the husk and seed?¿ How were the samples dried?
Line 157, brand, model, and country of water bath used. Minutes or min?
Line 158, brand, model, and country of the centrifuge. Minutes or min?
Line 159. How was it evaporated? Describe the conditions
Line 163. The standards ¿ where were obtained?
Line 169. Brand, model, and country of the oven. 0.2 g, do not usually start a sentence with a number.
Line 194. UV-Vis spectrophotometer, brand, model and country.
Line 252-255. The sentence is repetitive.

Experimental design

Regarding the determination of CFT it indicates that it is per 100 g of sample. And in total flavonoids, it indicates that it is per 100 g of dry extract....
So you are using the same sample to determine both? Or? please clarify this point. The same for the antioxidant activity assays.

Validity of the findings

Expand the discussion in the physicochemical and phytochemical parts since it only compares with other varieties but does not provide the reason for their results. If it affects the variety, the place where they were grown, etc.

Additional comments

Please restructure the conclusion based on what you report in your study.

---

## Round 0.2 · accepted · Accept

Thanks for addressing all the reviewers comments.

Reviewer 1 ·

Basic reporting

no comment

Experimental design

no comment

Validity of the findings

no comment

Reviewer 2 ·

Basic reporting

The authors have satisfactorily responded to the doubts, questions and comments made on the manuscript.

Experimental design

The authors have satisfactorily responded to the doubts, questions and comments made on the manuscript.

Validity of the findings

The authors have satisfactorily responded to the doubts, questions and comments made on the manuscript.

Additional comments

The authors have satisfactorily responded to the doubts, questions and comments made on the manuscript.